# Optimization of QuEChERS Method for Simultaneous Determination of Neonicotinoid Residues in Pollinator Forage

**DOI:** 10.3390/molecules25122732

**Published:** 2020-06-12

**Authors:** Maura J. Hall, Viet Dang, Steven P. Bradbury, Joel R. Coats

**Affiliations:** 1Department of Entomology, Iowa State University, 2007 ATRB, Iowa State University, Ames, IA 50011, USA; mjhall@iastate.edu (M.J.H.); spbrad@iastate.edu (S.P.B.); 2Department of Veterinary Diagnostic and Production Animal Medicine, Veterinary Diagnostic Laboratory, Analytical Chemistry Services Section, 1850 Christensen Drive, Iowa State University, Ames, IA 50011, USA; vdang.duc@gmail.com; 3Department Natural Resource Ecology and Management, 339 Science II, Iowa State University, Ames, IA 50011, USA

**Keywords:** neonicotinoid, insecticides, plant tissue, pollen, LC-MS/MS, method development, monitoring

## Abstract

Consistent with the large-scale use of pesticide seed treatments in U.S. field crop production, there has been an increased use of neonicotinoid-treated corn and soybean seed over the past decade. Neonicotinoids can move downwind to adjacent off-field pollinator habitats in dust from planting and/or move downslope to habitats in surface water. The extent of potential neonicotinoid exposure to pollinators from neonicotinoid movement into these adjacent pollinator habitats is unclear. Pollen and leaf tissue extractions were completed using a quick, easy, cheap, effective, rugged, and safe (QuEChERS) extraction procedure. Samples were subjected to a clean-up step using dispersive solid-phase extraction (dSPE) techniques prior to analysis. The compounds in the extracts were separated on a reversed-phase column with gradient elution and confirmed with tandem mass spectrometry. The extraction method showed acceptable recoveries of analytes ranging from 78.4 to 93.6% and 89.4 to 101% for leaf tissue and pollen, respectively. The method’s detection limits ranged from 0.04 to 0.3 ng/g in milkweed leaf tissue and 0.04 to 1.0 ng/g in pollen. The method is currently being employed in ongoing studies surveying pollen from a diversity of forbs and milkweed leaves obtained from habitat patches established within fields with a history of using neonicotinoid-treated seeds.

## 1. Introduction

Since neonicotinoids entered the market in the 1990s, they have become the fastest-growing class of insecticide worldwide [1,2]. Due to their wide-scale use as seed treatments, as well as foliar applications, neonicotinoids are now the most widely used class of insecticide in the world [3]. Their effectiveness against a broad spectrum of sucking and chewing pests and their unique mechanism of action have made them a commonly used group of insecticides in modern crop protection [4]. Neonicotinoids are synthetic compounds designed to act as agonists in the nicotinic acetylcholine receptors in the insects’ central nervous system, causing paralysis and death [5,6]. Imidacloprid, clothianidin, and thiamethoxam act systemically due to their relatively high water solubility (0.61, 0.34, and 0.41 g/L, respectively) [7]. Any insecticide that has widespread use can potentially have nontarget impacts on mammals, birds, and other vertebrates, as well as on nontarget insects and other invertebrates. The United State Environmental Protection Agency (U.S. EPA) does not consider there to be risks of concern for human health via dietary (i.e., food and drinking water consumption), residential, or bystander exposure to imidacloprid [8], clothianidin [9], or thiamethoxam [9]. Ecological risk assessments have been crucial for informing the registration of neonicotinoid insecticides. Potential toxicity to nontarget vertebrates has been summarized by Gibbons et al. [10] and Hladik et al. [6]. Neonicotinoids in surface water can also have impacts on invertebrates [6]. When formulated in seed treatments, these insecticides can be taken up by the roots of a plant and translocated throughout the stem, leaves, flowers, and pollen [5,6]. Studies have documented the presence of neonicotinoids in pollinator habitats; however, the extent to which exposures are within the range that produces detrimental effects in monarch butterfly larvae, honey bees, and native bees is unclear [11,12,13,14,15,16]. Efficient, multi-analyte residue analyses are needed to develop an accurate understanding of neonicotinoid exposure levels in order to achieve a better understanding of the potential effect of these insecticides on pollinators [5,6]. 

The objective of this project was to develop a fast and precise single extraction and analysis method for the three commonly used neonicotinoids (clothianidin, imidacloprid, and thiamethoxam) and two metabolites (5-OH-imidacloprid and imidacloprid olefin) in a pollinator-relevant matrix (see https://www.mzcloud.org for structures and fragmentation schemes). The goal of developing this method was allow for more effective exposure-monitoring studies to take place in pollinator habitats where potential exposure could occur. 

## 2. Results

### 2.1. UHPLC-MS/MS Method Optimization

To optimize multiple reaction monitoring (MRM) transitions for the individual compounds, standard solutions at 500 µg/L combined with 50:50 mobile phases A and B (1:1 *v*/*v*) were infused into the mass spectrometer at 10 µL/min. Each compound was examined under two different ionization techniques, ES+ and ES−, to achieve optimal sensitivity and selectivity. The best results were obtained using ES+ mode for parent compounds and two metabolites. Two MRM transitions were chosen for each analyte: one for quantitation and a second transition for confirmation. 

The two MRM transitions used for each analyte with the optimized mass spectrometry (MS) parameters are presented in Table 1. Several experiments were performed to evaluate chromatographic conditions, and better results were obtained with gradient elution settings with a flow rate of 300 µL/min. Neonicotinoids and metabolites were separated using the Accucore aQ column, with Retention Time (RT) ranging from 3.5 to 5.5 min. The typical MRM chromatograms of five compounds in spiked blank plant tissue are depicted in Figure 1.

### 2.2. Optimization of Sample Preparation

Clean-up of environmental samples is essential to minimize impairment of the analytical equipment and to eliminate matrix interference in the mass analyzer. Reliable clean-up is challenging with plant tissue and pollen samples due to the presence of pigments and lipids. Two major advancements for sample clean-up are the use of a quick, easy, cheap, effective, rugged, and safe (QuEChERS) procedure and dispersive solid phase extraction (dSPE). QuEChERS and dSPE are simple and robust techniques for the extraction and clean-up of analytes in different matrices, including biological tissues, food products, and environmental matrices [13,14,17]. 

#### 2.2.1. Plant Tissue Matrix 

To date, extraction of neonicotinoids from plant leaf tissues, specifically milkweed leaf tissue, is labor-intensive and involves complicated clean-up steps [16,18,19,20,21]. These methods use various components, such as Celite, C**_18_** cartridges, concentration steps, sodium chloride, anhydrous magnesium sulfate, filtration, and solvent exchange. By optimizing QuEChERS extraction and dSPE we have developed a method that reduces labor costs, variability, and the use of solvents [13,15,17,22]. 

To assess this approach with leaf tissue, we compared two commonly used QuEChERS methods. One method involved adding 5 g of milkweed powder into a 50 mL QuEChERS extraction tube containing 4 g MgSO_4_, 1 g NaCl, 1 g trisodium citrate dehydrate, and 0.5 g sodium citrate. The other method used 50 mL QuEChERS extraction tubes containing 4 g MgSO_4_ and 1 g NaCl. Samples, including blank and calibration standards, were spiked with internal standard mixture solution, and eight control samples were also spiked with an analyte mixture solution to make calibration standards. The samples were then extracted and cleaned up using dispersive solid-phase extraction (dSPE) containing 150 mg MgSO_4_, 25 mg primary secondary amine (PSA), and 7.5 mg graphitized carbon black (GCB) (data not shown). The extracts from each method were analyzed by liquid chromatography tandem mass spectrometry (LC-MS/MS). There was a clear correlation between a decrease in response for all analytes of interest when the 1 g trisodium citrate dehydrate and 0.5 g sodium citrate were not present. The 1 g trisodium citrate dehydrate and 0.5 g sodium citrate maintained the pH during the extraction, which improved the recovery. We thus moved forward using the QuEChERS extraction method of 4 g MgSO_4_, 1 g NaCl, 1 g trisodium citrate dehydrate, and 0.5 g sodium citrate followed by dSPE containing 150 mg MgSO_4_, 25 mg PSA, and 7.5 mg GCB. 

#### 2.2.2. Pollen Matrix 

Common practices for analyzing contaminates in pollen include the use of QuEChERS followed by clean-up using dSPE [14,17]. Most techniques require 1.0 g of pollen or more [17,22,23]. When working on field-level studies, it is often difficult to obtain 1.0 g of pollen. Hence, we adapted methods to quantify neonicotinoids in samples of 0.2 g or less. Given the low mass of our samples, we chose to use dSPE to remove complex compounds found within the pollen. Spiked pollen extract (1 mL) was transferred into a 1.5 mL micro-centrifuge tube containing 150 mg MgSO_4_, 25 mg PSA, and 7.5 mg GCB. The mixture was thoroughly vortexed for 1 min and centrifuged at 6500 rpm for 5 min. The same procedure was also performed with a 1.5 mL micro-centrifuge tube containing 150 mg MgSO_4_, 50 mg PSA, and 50 mg C**_18_**, as well as with a 1.5 mL micro-centrifuge tube containing 150 mg MgSO_4_, 50 mg PSA, 50 mg GCB, and 50 mg C**_18_**. In both dSPEs containing GCB, there was a substantial decrease in recovery for all analytes. However, the most significant decrease in recovery was for imidacloprid 5-hydroxy and imidacloprid olefin (up to 60%). We thus moved forward with the dSPE containing 150 mg MgSO_4_, 50 mg PSA, 50 mg GCB, and 50 mg C**_18,_** injecting 2 µL of the crude extract into the LC-MS/MS to minimize interfering compounds that can hamper method sensitivity. In addition, we used isotopically labeled internal standards for each of the analytes to correct for recovery throughout the extraction and analysis processes. 

### 2.3. Method Validation

Identification of the five analytes of interest was accomplished by comparing the retention time, peak shape, and ion ratio between solvent standards and sample spikes [24]. A total run time of 8 min was used for the separation of analytes (Figure 1). The performance of the LC-MS/MS method was validated using standard solutions spiked into control samples, sample blanks, and Quality Control (QC) samples. Linearity, matrix effects, method detection limit, precision, and recovery were examined.

#### 2.3.1. Evaluation of Linearity

Linearity for the five compounds was examined by analyzing eight calibration standards. Calibration curves were constructed by plotting the corresponding peak area ratios of analytes/internal standards against the concentration ratios of analytes/internal standards. The matrix-matched calibration curves obtained using simplified QuEChERS procedures were linear over the concentration range for the five analytes in pollen and plant tissue. Linearity, tested using the least-square regression method, gave a correlation coefficient (r^2^) greater than 0.980 in all the linear ranges. 

#### 2.3.2. Evaluation of the Method Detection Limit

Method detection limit (MDL) was estimated using the lowest concentration for which percentage relative standard deviation (%RSD) was less than or equal to 15%. Once that level had been determined, MDL was calculated using the formula S × 3.143, where S is the standard deviation of the calculated concentration among 7 replicates and 3.143 is the value for Student’s t-test for 6 degrees of freedom. The MDLs for the five analytes of interest ranged from 0.04 to 0.3 ng/g in milkweed leaf tissue (Table 2) and from 0.04 to 1.0 ng/g in pollen (Table 3). 

#### 2.3.3. Evaluation of Recovery

The recovery (extraction efficiency) was calculated by dividing the peak area of an analyte from a pre-extraction spiked sample by the peak area of an analyte from a post-extraction spiked sample. The extraction recoveries ranged from 85.4 to 93.6% for milkweed leaf tissue (Table 2) and from 89.4% to 101% for pollen (Table 3). 

#### 2.3.4. Evaluation of Trueness and Precision 

Intra-assay trueness [25] and precision were determined by analyzing three replicates of QC samples in a single LC-MS/MS run, while inter-assay trueness and precision were determined by analyzing four replicates of QC samples on two or more different days. The concentrations of QC samples were determined using calibration standards prepared on the same day. The assay trueness, presented as percentage, was calculated using the following equation: trueness = mean of calculated concentration/actual concentration × 100. The assay precision was determined by the relative standard deviation (%RSD) of the measured concentrations. The intra- and inter-day precision (%RSD) for the five analytes were within 20% of the reference values. The trueness of the method for milkweed leaf tissue ranged from 90.0 to 109% for the low QC level and from 78.4 to 103% for the high QC level (Table 2). The trueness of the method for pollen ranged from 93.6 to 111% for the low QC level and from 92.9 to 108% for the high QC level (Table 3).

## 3. Discussion

Our LC-MS/MS method is a sensitive, standardized, and labor-effective technique for analyzing pollen and milkweed leaf tissue, which are dietary sources for bees and monarch butterfly larvae, respectively. The proposed method allows for the quantification of the compounds in a single run at sub-ng/g concentrations using a faster and/or more sensitive method than those found in the literature [12,13,14,17,22,23,26,27,28]. The method can quantify specific analytes in a mixture of neonicotinoids at trace levels in small quantities of milkweed leaf and pollen to facilitate exposure assessment for honey bees, native bees, and monarch butterflies [12,18,19,29]. 

Pollinators are high-profile non-target organisms that may be exposed to neonicotinoids at levels of concern. The three active ingredients, clothianidin, imidacloprid, and thiamethoxam, are classified as highly toxic to bees, while little is known about their toxicity when monarch larvae are exposed to them [6,29]. Milkweed plant tissue and pollen taken from plants and bees located in close proximity to crop fields that are known to have had neonicotinoid seed treatment are key matrices to evaluate pollinator exposure levels. To accurately, precisely, and efficiently measure these levels, a robust multi-analyte method with a low MDL is needed. The method reported here is being used to analyze 500 plant tissue samples and 600 pollen samples. The adaption of these methods into other laboratories can help support the standardization of analytical techniques used to consistently evaluate neonicotinoid exposure for non-target organisms across research efforts.

## 4. Materials and Methods 

### 4.1. Standards, Reagents, and Solvents

A neat standard of imidacloprid (CAS 138261-41-3, 98.8% pure), thiamethoxam (CAS 153719-23-4, 95.2% pure), clothianidin (CAS 210880-92-5, 99.6% pure), imidacloprid-olefin (CAS 115086-54-9, 97.9% pure), and 5-OH-imidacloprid (CAS 380912-09-4, 96.7% pure) were received as a gift from Bayer CropScience (Research Triangle Park, NC, USA). Deuterated internal standards clothianidin-d3 and thiamethoxam-d3 were obtained from Sigma-Aldrich (St. Louis, MO, USA). Imidacloprid olefin-**^13^**C3,**^15^**N and imidacloprid-pyr-d_4_-methyl-d_2_, **^13^**C were received from Bayer CropScience, and 5-OH-imidacloprid-**^13^**C,**^15^**N was received from Clearsynth (Mississauga, Ontario, Canada). Organic solvents (Optima LC-MS grade methanol, water, and acetonitrile), ammonium formate (99% pure), and 99% pure formic acid were purchased from Fisher Scientific (Fair Lawn, NJ, USA). Fifty-milliliter QuEChERS tubes (part number 60105-216) and dispersive solid-phase extraction (dSPE) equipment (part number 60105-202; 60105-222; 60105-223) were obtained from Thermo Fisher Scientific.

### 4.2. Spiking Solution Preparation

Stock solutions (0.5 mg/mL) of individual standards and internal standards were prepared by dissolving 5 mg (corrected for salt and purity) in 10 mL solvent (e.g., acetonitrile, dimethylformamide, methanol, or dimethyl sulfoxide). Dilutions of the stock solutions were prepared in acetonitrile for spiking pollen (0.005 to 0.05 ng/µL) and leaf tissue (0.2 ng/µL). Internal standard solutions were prepared in acetonitrile at a concentration of 0.4 ng/µL. Working solutions of analytes and internal standards were stored at −200 °C and were freshly prepared monthly. 

### 4.3. Leaf Sample Preparation

Unexposed “control” common milkweed (*Asclepias syriaca*) leaves were obtained from Iowa State University (Ames, IA, USA) greenhouses with no history of neonicotinoid use (as noted in the method validation, levels were below the method detection limit). Leaf samples were stored at −80 °C prior to extraction. On the day of extraction, samples were pulverized using a blender with a small amount of dry ice. The leaf powder was then placed in a fume hood to sublimate the remaining dry ice. The leaf powder was extracted following a generic QuEChERS method with some modifications [13]. In brief, approximately 5 g of powder was weighed into a 50 mL QuEChERS extraction tube (Thermo Fisher, catalog number 60105-216) containing 4 g MgSO_4_, 1 g NaCl, 1 g trisodium citrate dehydrate, and 0.5 g sodium citrate. Samples, including blank, QC levels, and calibration standards, were spiked with 100 µL of an internal standard mixture solution (8 ng/g sample). Eight control samples were also spiked with an analyte mixture solution (0.2 ng/µL) to make calibration standards of 0.2, 0.5, 1, 2, 5, 10, 20, and 40 ng/g samples. Two QC levels (low = 1 ng/g and high = 20 ng/g) in triplicates were also included. Samples were solvent extracted with 10 mL of LC-MS-grade acetonitrile, followed by vortexing for 30 s and shaking on a multi-tube shaker for 10 min at 2500 rpm. The samples were then centrifuged for 6 min at 3700 rpm. After centrifugation, 1 mL of supernatant was transferred into a 2 mL dSPE tube (Thermo Fisher, catalog number 60105-222) containing 150 mg MgSO**_4_**, 25 mg primary secondary amine (PSA), and 7.5 mg graphitized carbon black (GCB), after which it was vortexed for 1 min. The sample tubes were subsequently centrifuged for 5 min at 5000 rpm. The supernatants (~700 µL) were filtered using a 0.45 µm filter and transferred into amber autosampler vials prior to LC-MS analysis.

### 4.4. Pollen Sample Preparation 

Pesticide-free pollen (Buzzy Bee; purchased from Amazon) was analyzed for background levels of neonicotinoids before the spiking tests, and it was found that levels were below the method detection limits. The milkweed leaf sample extraction method was modified for low-mass pollen samples collected during monitoring studies. In brief, approximately 0.2 g of pollen was weighed into a 2 mL prefilled tube kit containing high impact zirconium beads of 1.5 mm diameter (Benchmark, catalog number D1032-15). 

Samples were extracted with 0.3 mL of water and then shaken on a multi-tube shaker for 5 min at 2500 rpm. Acetonitrile (1.2 mL) was then added to all samples, followed by shaking on a multi-tube shaker for 5 min at 2500 rpm. The samples were then centrifuged for 5 min at 6000 rpm. After centrifugation, 1 mL of supernatant was transferred into a 2 mL dSPE tube (Fisher Scientific, catalog number 03150625) containing 150 mg MgSO_4_, 50 mg PSA, and 50 mg C**_18_** and then shaken on a multi-tube shaker for 2 min at 2500 rpm. The sample tubes were centrifuged for 5 min at 6000 rpm. The supernatant (300 µL) was transferred into an amber autosampler vial with an insert prior to LC-MS analysis. 

### 4.5. LC-MS Conditions

The LC-MS/MS consisted of a Vanquish Flex UHPLC system, including a binary pump, autosampler, and column heater compartment, and a TSQ Altis triple quadrupole mass spectrometer equipped with heated electrospray source (Thermo Fisher Scientific, San Jose, CA).

Chromatographic separation was carried out on an Accucore aQ column (100 × 2.1 mm, 2.6 µm; Thermo Fisher Scientific). The column was maintained at 30 °C. The mobile phase consisted of water:methanol (95:5 *v*/*v*) containing 0.1% formic acid and 5 mM ammonium formate (A) and methanol:water (95:5 *v*/*v*) containing 0.1% formic acid and 5 mM ammonium formate (B). The elution gradient was held at 0% B for the first 0.5 min, increased from 0 to 80% B from 0.5 to 6 min, held at 80% B from 6 to 8 min, decreased from 80% to 0% B from 8 to 9 min, and held at 0% B for 1 min. The flow rate was 0.3 mL/min for the duration of the run. Injection volume was 2 μL. The needle wash was a mixture of water:methanol (80:20 *v*/*v*).

The MS ionization source conditions were optimized via direct infusion of standard solutions into the mass spectrometer. The mass spectrometer was operated in positive ion heated electrospray ionization mode. The electrospray voltage was set at 3700 V for positive mode. Nitrogen was used as a sheath gas (30 arb), auxiliary gas (6 arb), and sweep gas (1 arb). Argon was used as a collision gas. Ion transfer tube and vaporizer temperatures were set at 325 and 350 °C, respectively. Acquisition was performed in selected reaction monitoring (SRM) mode, and two or three main transitions were monitored for each compound (supporting information can be found in Table 1). Data analysis was performed on TraceFinder 4.1 software (Thermo Fisher Scientific,).

## 5. Conclusions

We were able to develop a single extraction and quantitation method for a suite of neonicotinoids that are commonly used as seed treatments for corn and soybean (clothianidin, imidacloprid, and thiamethoxam) and two imidacloprid metabolites (5-OH-imidacloprid and imidacloprid olefin) in a pollinator-relevant matrix. Analysis of leaf tissue and pollen is essential to allow for reasonable estimates of exposure for monarch larvae and bees. Since collection of these samples is resource intensive, it is critical to develop efficient and accurate extraction and quantification methods. Currently, sample preparation in combination with LC-MS/MS for neonicotinoid quantification in plant tissue and pollen has been limited to more intensive extraction methods with high-mass samples and longer LC-MS/MS runs [14,16,17,18,19,22,23,30]. To address these limitations, we developed a single extraction and analytical method for multiple neonicotinoids from milkweed leaf tissue and pollen. Our method’s performance is comparable to, and in some cases superior to, existing methods. The method will support more cost-effective monitoring studies that improve understanding of the spatiotemporal variation of these compounds within agro-ecostyems. The method we report could be evaluated and adapted as needed to support the quantification of multiple neonicotinoid concentrations in animal tissues and other environmental matrices relevant to human health, aquatic life, and wildlife risk assessments. 

## Figures and Tables

**Figure 1 molecules-25-02732-f001:**
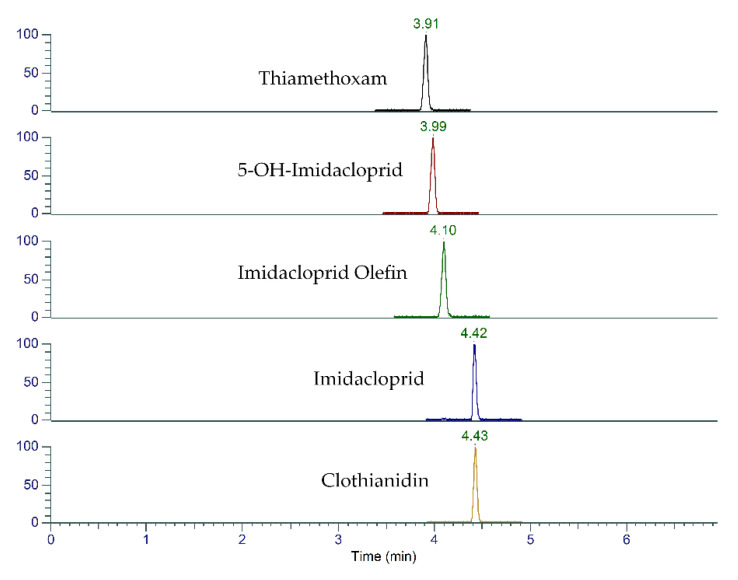
The liquid chromatography tandem mass spectrometry (LC-MS/MS) chromatogram of five neonicotinoids spiked into milkweed leaf tissue extract at 40 ng/g. Across the chromatographic peak, 10 to 12 data points were obtained.

**Table 1 molecules-25-02732-t001:** Select reaction monitoring (SRM) table of three neonicotinoids and their metabolites**.**

Compound	Retention Time (min)	Precursor (*m*/*z*)	Product (*m*/*z*)	Collision Energy (V)	RF Lens (V)
Imidacloprid olefin	4.08	254	171	17.2	46
Imidacloprid olefin	4.08	254	205	13.8	46
Imidacloprid	4.41	256	175	19.5	41
Imidacloprid	4.41	256	209	15.4	41
Imidacloprid	4.41	256	212	10.2	41
Clothianidin	4.42	250	113	26.2	34
Clothianidin	4.42	250	169	14.3	34
Thiamethoxam	3.88	292	132	21.9	34
Thiamethoxam	3.88	292	181	22.3	34
Thiamethoxam	3.88	292	211	10.2	34
5-OH-imidacloprid	3.96	272	134	41.2	53
5-OH-imidacloprid	3.96	272	191	19.1	53
5-OH-imidacloprid	3.96	272	225	15.0	53
Imidacloprid olefin-13C_3_,15N	4.08	259	176	17.2	43
Imidacloprid olefin-13C_3_,15N	4.08	259	211	10.2	43
Imidacloprid olefin-13C_3_,15N	4.08	259	241	10.2	43
Imidacloprid-pyr-d4-CH3-d2,^13^C	4.39	260	179	18.6	47
Imidacloprid-pyr-d4-CH3-d2,^13^C	4.39	260	213	16.1	47
Imidacloprid-pyr-d4-CH3-d2,^13^C	4.39	260	214	10.5	47
Clothianidin-d3	4.41	253	132	16.6	44
Clothianidin-d3	4.41	253	172	12.7	44
Thiamethoxam-d3	3.88	295	184	22.4	45
Thiamethoxam-d3	3.88	295	214	11.7	45
5-OH-imidacloprid-d4	3.95	276	195	19.7	51
5-OH-imidacloprid-d4	3.95	276	229	13.3	51

**Table 2 molecules-25-02732-t002:** Method detection limit (MDL), recovery, trueness, and matrix effect for five target analytes in a common milkweed leaf matrix**.**

			Trueness	
Compound ID	MDL (ng/g)	Recovery (%)	Low QC	High QC
Thiamethoxam	0.04	93.6	90.1	100
5-OH-imidacloprid	0.1	78.4	96.6	78.4
Imidacloprid olefin	0.1	90.2	109	98.1
Imidacloprid	0.2	86.7	90.0	103
Clothianidin	0.3	85.4	90.0	100

**Table 3 molecules-25-02732-t003:** Method detection limit (MDL), recovery, trueness, and matrix effect for five target analytes in the pollen matrix**.**

			Trueness	
Compound ID	MDL (ng/g)	Recovery (%)	Low QC	High QC
Thiamethoxam	0.04	101	99.7	95.5
5-OH-imidacloprid	0.3	89.4	95.8	100
Imidacloprid olefin	1.0	94.5	111	108
Imidacloprid	0.06	99.7	93.6	95.0
Clothianidin	0.06	95.8	96.0	92.9

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
