# Peer review of "Optimization of QuEChERS Method for Simultaneous Determination of Neonicotinoid Residues in Pollinator Forage"

_molecules, 2020, doi:10.3390/molecules25122732_

Round 1
Reviewer 1 Report
For AuthorsStrength and weakness of this paper, so that authors can make response:
In General:
Conclusions
Instructions for Authors Says: This section is not mandatory, but can be added to the manuscript if the discussion is unusually long or complex
I think the work should improve the discussion and should include a conclusion with the relevant results of your research in general
Specific comments
In the Introduction I suggest to include:
Point 1: From a reviwers perspective, is entirely clear what the science objective of the article principally at the end of introduction (bees and pollinators). There are several areas where this paper could be improved. “The introduction would benefit from brief comments on the general Known human health hazards and risks, Contamination of the Aquatic Environment with Neonicotinoids and its Implication for Ecosystems (include lethal dose, LD50). However, the authors should point out that this pesticide is highly toxic in vertebrate wildlife—mammals, birds, amphibians and reptiles and that risk mitigation measures should be considered to make its introduction more complete. I think this needs to be spelt out more clearly and promoted more strongly in the article. Point 1.1 authors should include this reference GIBBONS, David; MORRISSEY, Christy; MINEAU, Pierre. A review of the direct and indirect effects of neonicotinoids and fipronil on vertebrate wildlife. Environmental Science and Pollution Research, 2015, vol. 22, no 1, p. 103-118.
Point 2 Results: Since this is a journal of chemistry, it would be didactic to include a Structure and fragmentation scheme for pesticides in results. (Mechanism of fragmentation in mass spectra).
Point 2.1 line 195 authors write 0.05 ng/µl) and should be (0.05 ng/µL)
Point 3.0 Discussion: The authors should also discuss effects of the subchronic exposure to environmental concentrations of nicotinoids in freshwater gastropod.
Point 3.1 Disccusion: This is an analytical chemistry article, and the results and discussions of this topic are good in my opinion . But it should be enriched by what it produces in health and environment, so I suggest discussing imidacloprid. Imidacloprid is can lead to testicular anomalies, DNA damage in males. should be included mechanism action of this pesticide.
Point 4 Discussion:
Specifically discuss the pollen's ability to absorb neonicotinoids because it is reported that bee pollen absorbs pollution from the environment, which may include pesticides.
Point 5 Discussion: Authors should also discuss stadistical results and indicate the precision and accuracy of your new method.
Point 6
Line 51-54 : Authors says to address these limitations, we have developed a single extraction and analytical method for multiple neonicotinoids from milkweed leaf tissue and pollen. Our method’s performance is comparable to and in some cases superior to existing methods.this comment should be reflected in conclusions (should be included) because it must be clear that there is a substantial improvement in the methodology developed by these researchers
Reviewer 2 Report
Minor revision is required before this paper can be recommended for acceptance for publication.
Lines 48-54: this paragraph is confusing and needs to be rewritten. The objective of the study should be explained better.
Lines 57: MRM?? Specify, please
Line 92: "dispersive solid-phase extraciton (dSPE)" should be specify the first time it comes into view in the text, that is in line 77, as well as the QuEChERS word.
A conclusion section should be done.
